# A Management Method of Multi-Granularity Dimensions for Spatiotemporal Data

Wen Cao [1,*], Wenhao Liu [1], Xiaochong Tong [2], Jianfei Wang [1], Feilin Peng [3], Yuzhen Tian [1] and Jingwen Zhu [1]

1    School of Geoscience and Technology, Zhengzhou University, Zhengzhou 450001, China
2    School of Geospatial Information, Information Engineering University, Zhengzhou 450001, China
3    Zhongke Yungu Technology Co., Ltd., Changsha 410000, China
*    Correspondence: zzdx_edifier@zzu.edu.cn

**Abstract:** To understand the complex phenomena in social space and monitor the dynamic changes in people's tracks, we need more cross-scale data. However, when we retrieve data, we often ignore the impact of multi-scale, resulting in incomplete results. To solve this problem, we proposed a management method of multi-granularity dimensions for spatiotemporal data. This method systematically described dimension granularity and the fuzzy caused by dimension granularity, and used multi-scale integer coding technology to organize and manage multi-granularity dimensions, and realized the integrity of the data query results according to the correlation between the different scale codes. We simulated the time and band data for the experiment. The experimental results showed that: (1) this method effectively solves the problem of incomplete query results of the intersection query method. (2) Compared with traditional string encoding, the query efficiency of multiscale integer encoding is twice as high. (3) The proportion of different dimension granularity has an impact on the query effect of multi-scale integer coding. When the proportion of fine-grained data is high, the advantage of multi-scale integer coding is greater.

**Keywords:** spatiotemporal big data; dimensions; dimension granularity fuzziness; multi-scale integer coding

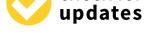



## 1. Introduction

The spatiotemporal data reflect the quantitative and qualitative characteristics, spatial structure, spatial relations, and their changes with time of various elements or phenomena in the geographical world, which is the basis for human cognition of the geographical world. Most of the problems we face must be addressed based on data-driven approaches for understanding better and achieving more efficient and optimal decisions [1,2]. In recent years, with the development of Internet technology and sensor equipment, the production mode of spatiotemporal data has changed from passive production and active production to automatic production, which makes the spatiotemporal data resources we obtain more abundant [3–5]. We can use the monitoring video to analyze the vehicle operation and the movement rule of people on the road, realize the tracking and real-time prediction of traffic conditions, and avoid traffic and congestion. It can also improve the accuracy of weather forecasting by establishing models for continuous years of observation data. However, many social phenomena are complex. In order to reveal their essence, we need more cross-scale data [6,7]. For example, in recent years, the outbreak of COVID-19 has seriously affected the development of society. Many scholars have contributed to the prevention of the outbreak of COVID-19 by analyzing the correlation between relevant indicators at different scales and confirmed cases of COVID-19 [8,9]. Therefore, we can see the importance of different scale data for data mining. However, when we query data, we often ignore the multi-scale impact of data, resulting in incomplete data acquisition. Therefore, there is an urgent need for a multi-granularity dimension management method

of spatiotemporal data. The rest of this paper was organized as follows: "Related work" introduced the content related to this study. "Method" introduced the DGFQM and the multi-scale dimension integer coding method. "Results" verified the effectiveness of the method and analyzed its results. "Discussion" was the content related to the results, including shortcomings and prospects.

Multi-granularity dimensions of spatiotemporal data mainly included time and space. First, we understand how to manage time and space in spatiotemporal data management. A popular direction of spatial information management was the method based on the grid model [10,11], which used a spatial filling curve to build grid code, and improved the indexing and query efficiency of multi-scale spatial data. For example, Guo [12] and others proposed an adaptive Hilbert –Geohash geogrid and coding method. Cao et al. [13] and others used a Hilbert curve to store and retrieve spatiotemporal data. However, geohash encoding lacked cross-scale spatial relations, resulting in low indexing efficiency. Zhai et al. [14] and others proposed a level-by-level space-filling curve, which improves the correlation between multiple levels by connecting adjacent levels. The clustering between levels of this method was poor, and the existing spatial retrieval strategy only considered the intersection and the included data that was not considered [15]. To solve the above problems, Lei et al. [16] proposed a global multi-scale spatial grid coding model, and designed a strategy to ensure the integrity of spatial queries based on this model. Multi-scale time was more used for auxiliary processing in simple ways, such as timestamps [17] and strings [18], which were scattered in file systems [19–21], databases [22–24], and programming languages. This time management method did not retain the multi-scale information of time, and it was difficult to manage the multi-scale time uniformly. To solve this problem, Tong et al. [25] proposed a multi-scale time segment integer coding method, which uses integer representation of time scale information and location information. However, the management of multi-granularity dimensions had the following challenges:

First, the intersection query results based on time and other one-dimensional dimensions are incomplete, and the impact of time multi-scale is not considered. The current fuzzy query method used fuzzy set theory [26] to solve the problem of fuzzy words such as "left and right" and "probably" in time description, and could solve the new fuzzy problem caused by multi-granularity [27,28].

Secondly, the current research on multi-granularity dimensions was mainly about time and space, and other multi-scale dimensions were not discussed. As a kind of spatiotemporal data, remote sensing data have the ability to cover a wide area of spectrum. The spectral band spanned from visible light, thermal infrared to microwave, and the resolution changed from multispectral to hyperspectral, which was an important indicator for distinguishing physical properties of ground objects in remote sensing data [29,30]. Recently, the application of radio wave imaging technology in daily life had made the band information span larger [31–33]. Zhang et al. [34] designed five spatiotemporal spectral integrated storage formats for long-term remote sensing data with time, space, and spectral information. However, there were few studies on multiscale bands. At present, band information was represented by unique identifiers in the database system. This method is not conducive to the unified storage of multi-source information.

Based on the above analysis, we propose a multi-granularity dimensions management method for spatiotemporal data, which is mainly divided into the DGFQM and multi-scale dimension integer coding. The DGFQM divided the query results into fuzzy data and fine data according to the dimension granularities, and obtained complete query results according to the correlation between different scale codes. Multi-scale dimension integer coding mainly applied the multi-scale integer coding method to the band. We designed an association method between arbitrary scale band and multi-scale integer coding to improve the efficiency of data retrieval.

## 2. Materials and Methods

### 2.1. Dimension

Dimension refers to the inherent and measurable physical properties of physical quantities. Internationally, the seven basic dimensions, such as time and length, are often used to represent other physical quantities. Physical quantity under the same dimension has the function of ordering and is often used to retrieve conditions. However, the dimensions have multi-granularity characteristics. The multi-granularity dimensions easily lead to data loss, so we defined the concept of dimension granularity fuzziness, and described the fuzziness problem.

#### 2.1.1. Dimension Granularity

Granularity refers to the size of the particles. Granularity is measured by particle diameter (usually long or medium diameter). We expressed the measure of physical quantity with dimension granularities. Database systems usually use existing units to represent dimension granularities, such as standard time units (year, month, day, etc.), and length units (meter, decimeter, millimeter, etc.). The premise of realizing this goal requires a simple, effective, and easy-to-use multi-granularity dimension system. Therefore, we defined the relevant concepts as follows:

**Definition 1.** *Dimension Domain, D is a set of completely ordered points that satisfy a sorting relation. $D = \{d_1, d_2, d_3, \ldots, d_n\}$, where $d_1 < d_2 < d_3 < \ldots < d_n$.*

**Definition 2.** *Dimension Particle, G is a set of finite continuous points in a dimension domain. $G = \{d_1, d_2, \ldots, d_k\}$, where k is the number of aggregate particles.*

**Definition 3.** *Dimension Granularity, R is a set of nonoverlapping dimension particles, $R = \{G_1, G_2, G_3, \ldots, G_n\}$.*

**Definition 4.** *Dimension Granularity Relations refer to the correlation between different dimension granularities. The relationship between Dimension Granularities can be divided into equal relations, finer relations, and coarser relations according to the size of the particles that make up the Dimension Granularity. Assume that $R_1$ and $R_2$ are two different granularities, and $k_1$ and $k_2$ are the numbers of particles contained in $R_1$ and $R_2$. $k_1 = k_2$, the granularity of $R_1$ is equal to that of $R_2$, $R_1 = R_2$; $k_1 < k_2$, the granularity of $R_1$ is smaller than that of $R_2$, $R_1 \nleq R_2$; $k_1 > k_2$, the granularity of $R_1$ is larger than that of $R_2$, $R_1 \ngtr R_2$.*

**Definition 5.** *Inclusion Relation, A point at a certain granularity can be expressed as a set of finite points of another granularity, and the inclusion relation exists between the two granularities. Assume that $R_1$ and $R_2$ are two different granularities, $R_1 \nleq R_2$. For any point at the granularity of $R_2$, there is always a finite number of corresponding points at the granularity of $R_1$, where $x_1$ is the point at the granularity of $R_2$ and $y_i$ is the point at the granularity of $R_1$.*

There are specific conversion rules between these units such as 60 min for an hour. However, there is not only fixed granularity information but also other granularity information. Therefore, it is urgent to implement the limited granularities to represent other granularities. The dimensions have two different representations: point and segment type. The point type represents a position on a dimension domain, represented by a value of a certain granularity. The segment type represents the interval on the dimension domain, which is represented by two points. This representation method realizes the representation of various granularities by existing units.

#### 2.1.2. Dimension Granularity Fuzziness

At present, the fuzzy problem adopts the fuzzy set theory. The method calculates the probability of fuzzy data occurrence through the membership function. In this way, the

fuzzy point was represented as a two-tuple $(d_1, \delta_1)$, where $d_1$ represented the point, and $\delta_1$ represented the membership degree. The fuzzy segment was converted into a quad-tuple $(d_1, \delta_1, d_2, \delta_2)$, where $d_1$ and $d_2$ were the start point and endpoint, and $\delta_1$ and $\delta_2$ were the membership degrees of the start point and endpoint, respectively. The premise of using the fuzzy set theory was to obtain the fuzzy data set. However, the fuzzy data sets were obtained through semantic computing or empirical knowledge. The above methods cannot solve the fuzziness caused by multi-granularity dimensions. Therefore, we described point fuzziness and segment fuzziness separately.

Compared with fine-grained data, coarse-grained data with multi-granularity dimensions has uncertainty. Therefore, different granularity choices for the same event produce different results. We defined the fuzziness induced by multi-granularity as the dimension granularity fuzziness, describing the fuzziness problem of point type and segment type, respectively.

Point

At present, most database systems use a point of a certain granularity to represent the state of an object, which is usually an index value. There are different granularities in practical applications, so granularity conversion is needed. We defined the granularity transformation function $T$:

$$T(d_R, H) = h, \tag{1}$$

where $d_R$ is a point at the granularity of $R$, $H$ is a granularity of transformation, and $h$ is a point at the granularity of $H$.

Assume that $d_1$ is a point at the granularity of $R_1$, and $R_2$ is a different granularity from $R_1$. The conversion of $d_1$ from the granularity of $R_1$ to that of $R_2$ involves the following two cases: $R_1 \not< R_2$, there is a unique dimension point $d_2$ at the granularity of $R_2$, i.e., $d_2 = h$; $R_1 \not> R_2$, $\{d_2 \mid l < d_2 < u\} = h$, where $l \sim u$ is a point set of $R_2$. The constant is generally used as the retrieval condition, so we divided the transfer function $T$ into $T_s$ and $T_l$.

$$T_s = min(T(d_R, H)), \tag{2}$$

$$T_l = max(T(d_R, H)), \tag{3}$$

where $T_s$ is the minimum value converted to $H$ granularity, $T_l$ is the maximum value converted to $H$ granularity.

Because of the multi-granularity characteristic of dimension, different granularity description of the same event produces different results. When describing the same event, coarse-grained points are fuzzier than fine-grained points. For example, the Wenchuan earthquake occurred on 12 May 2018 (China Standard Time), and the time under the annual granularity is 2018. The time information at annual granularity is fuzzier than that at daily granularity. We may miss this fuzzy information when retrieving data.

Segment

The segment represents a binary group $[d_1, d_2]$, which is all points between $d_1$ and $d_2$. The granularity of $d_1$ and $d_2$ are $R_1$ and $R_2$. In the ideal case, the segment can represent by one index value. However, the length of the segment does not correspond to the existing granularity. Currently, dimension segments represent by two fields, which is inefficient when querying. With the introduction of multi-scale integer coding, we designed the following rules to attain a reasonable and smaller number of index values to represent segments. According to the granularity relationship between $d_1$ and $d_2$, there are two kinds of cases.

Case 1: The granularity of $d_1$ is equal to $d_2$, i.e., $R_1 = R_2$. Assuming the interval length is $L$. If $R_x = L$, select the value at the granularity of $R_x$ to represent this interval, as shown in Figure 1a. If $R_x \neq L$, there are two filling methods. One is to fill from coarse-grained to fine-grained. The following three situations may exist depending on the coverage position of the index value. (1) As shown in Figure 1b, the index value covers the middle position

of the interval. (2) As shown in Figure 1c, the coverage of the index value starts at the starting position $d_1$. (3) As shown in Figure 1d, the coverage of the index value ends at the endpoint $d_2$. Recalculate the length of the remaining sections and repeat the above steps until all sections $[d_1, d_2]$ are covered. The other is to fill from fine-grained to coarse-grained. We can choose to start filling from the start point $d_1$ or the endpoint $d_2$. This method only needs to determine the granularity of the starting index value and does not need to perform multiple calculations. Therefore, we choose this method to study the band.

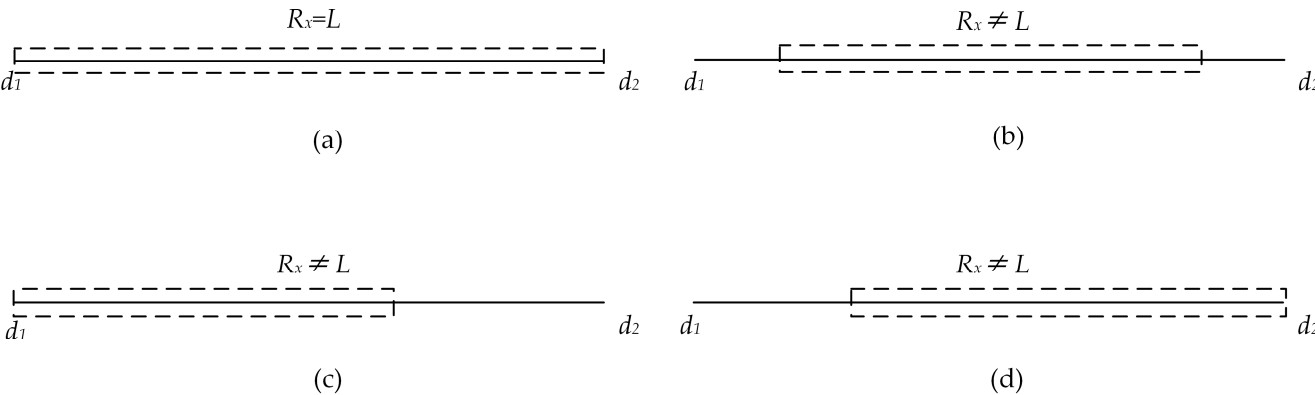

**Figure 1.** Different cases of building the index from coarse to fine-grained. (**a**) the index value covers the middle position of the interval. (**b**) the index value covers the middle position of the interval. (**c**) the coverage of the index value starts at the starting position $d_1$. (**d**) the coverage of the index value ends at the endpoint $d_2$.

Case 2: The granularity of $d_1$ is not equal to $d_2$, $R_1 \neq R_2$. First, we needed to convert the coarse-grained point to a fine-grained point. If $R_1 \ngtr R_2$, we reached the point of $d_1$ at $R_2$ granularity by the transformation function Ts. If $R_1 \nless R_2$, we converted d2 to the point at $R_1$ granularity through the transformation function $T_l$. According to the transformation function, the starting and ending points of the segment have the same granularity. Secondly, design the index values according to case 1. The fuzzy problem of segment type is similar to that of point type. Let the segment $D$ consists of several segments, $D= \{D_1, D_2, \dots, D_n\}$. T is fuzzy relative to $D_i$.

*2.2. DGFQM*

There are two main ways to retrieve data through dimensions. One is to query through a point, and the other is to query through the start point and end point, also known as the intersection query. Due to dimension granularity fuzziness, data are easily lost when querying, such as in the following example:

Data record 1: MODIS blue-band image (450–530 nm) of Beiyuan Road, Chaoyang District, Beijing, at 14:00 am on 15 November 2014.

Data record 2: MODIS visible-band image (380–780 nm) of Chaoyang District, Beijing, 15 November 2014.

Data record 3: MODIS panchromatic image (350–900 nm), Beijing, November 2014.

The above examples show that the same data was described differently due to the multi-granularity characteristics of temporal, spatial, and spectral attributes. Data record 1 was more accurate than data record 2, and data record 3 was fuzzier than data record 2. Important data may be missing from query results.

There are two kinds of missing data caused by dimension granularity fuzziness: coarse-grained missing and fine-grained missing data. Therefore, we divided the query results into fuzzy and exact data according to scales. Assume O ($p_1, p_2, \dots, p_i$) is an object with multiple attributes, where $p_i$ represents the $i$-th attribute. Take the intersection query as an example. Let the query interval be $[p_i^1, p_i^2]$, the corresponding scales are $N_1$

and $N_2$, respectively. We divided the query result $S$ into $S_1$, $S_2$, and $S_3$, $S_1$, $S_2$, and $S_3$, i.e., $S = S_1 \cup S_2 \cup S_3$.

$$s_1 = \left\{ O \middle| O(p_i) > \max(O(p_i^1), O(p_i^2)) \right\} \tag{4}$$

$$s_2 = \left\{ O \middle| \min(O(p_i^1), O(p_i^2) \le O(p_i^1) \le \max(O(p_i^1), O(p_i^2) \right\} \tag{5}$$

$$s_3 = \left\{ O \middle| O(p_i) < \min(O(p_i^1), O(p_i^2)) \right\} \tag{6}$$

where $S_1$ is the set of objects whose scales are larger than $p_i^1$ and $p_i^2$; $S_2$ is the set of objects whose scales are between $N_1$ and $N_2$; and $S_3$ is the set of objects whose scales are smaller than $p_i^1$ and $p_i^2$.

When $N_1 = N_2$, $S_1$ is the fuzzy data set and $S_2$ and $S_3$ are the exact data set. When $N_1 \ne N_2$, $S_1$ and $S_2$ are the fuzzy data set and $S_3$ is the exact data set. The DGFQM is to obtain the missing accurate data and fuzzy data. This method obtains missing data by analyzing the relationship between different dimension granularity. Since the specific steps of this method are related to the dimension coding method, we will introduce them in Section 3.

In practical application, the dimension granularity fuzzy query method must satisfy the following conditions:

Condition 1: The dimension has a multi-scale characteristic in the concrete application.

Condition 2: Inclusion relationships exist between adjacent levels.

Condition 1 means that a dimension domain can be represented by sets of points with different dimension granularities, or a point can be represented by multiple granularities. Condition 2 means that there is an inclusion relationship between adjacent levels, and a point on a certain scale includes all points on the next fine scale. As shown in Figure 2a, $d_1$ is expressed as one index value at $R_1$ granularity, two index values at $R_2$ granularity, and three index values at $R_3$ granularity, respectively. However, $R_2$ and $R_3$ do not satisfy condition 2. As shown in Figure 2b, $d_1$ can be represented as one index value at $R_1$ granularity, two index values at $R_2$ granularity, and four index values at $R_3$ granularity. Therefore, there is an inclusion relationship between adjacent granularities, which satisfies condition 2.

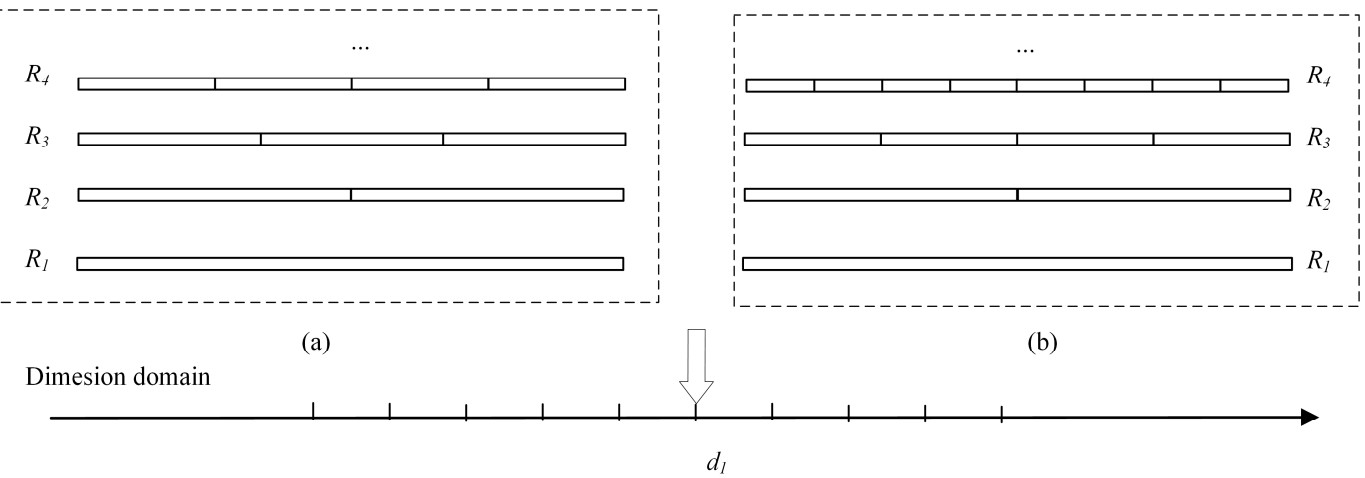

**Figure 2.** Different representations of points on dimension domains. (**a**) $d_1$ is represented as one index value at $R_1$ granularity, two index values at $R_2$ granularity, and three index values at $R_3$ granularity, respectively. (**b**) $d_1$ is represented as one index value at $R_1$ granularity, two index values at $R_2$ granularity, and four index values at $R_3$ granularity.

### 2.3. Dimension Coding Method

At present, dimensions are expressed in two ways: single-scale dimension coding and multi-scale dimension coding. Single-scale dimension coding is the representation of multi-granularity dimensions on a fixed scale. Multi-scale dimension coding represents multi-granularity dimension by coding at different scales. The existing coding methods are string coding and multi-scale integer coding. The multi-scale integer coding had been used in the time segment (multi-scale time segment integer encoding, MTSIC). For a time, MTSIC has had certain advantages compared to string coding. We extended it to multi-granularity dimensions, and the implementation method was as follows:

Assuming the dimension is $\dim(\alpha_1, \alpha_2 \dots, \alpha_{n-1}, \alpha_n)$, where $\alpha_i$ is the number of dimension components and $n$ is the number of dimension components. Figure 3 shows the principle of multi-scale dimension integer coding. Firstly, the components of the dimension are expressed in binary, and the single-scale dimension integer coding is formed by bit operation. Then, the multi-scale dimension integer coding is obtained based on the level information $N$. Since the bands usually exist in the form of dimension segments, we used multi-scale integer coding to manage the bands and designed the association method between multi-scale integer coding and band.

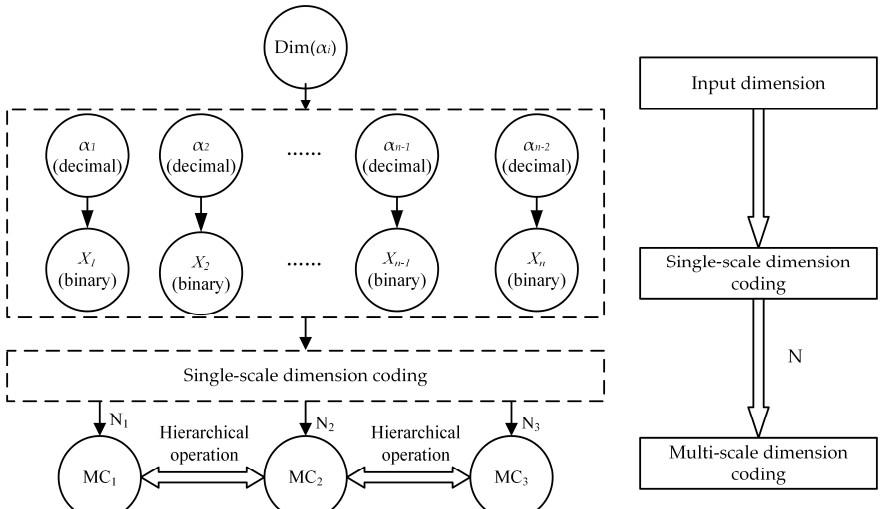

**Figure 3.** The principle of multi-scale dimension integer coding.

### 2.3.1. Multi-Scale Band Integer Coding

The band is encoded with an integer for single-scale band integer coding (SBIC) and multi-scale band integer coding (MBIC). The main idea of MBIC is to transform the band information into an SBIC, and then transform the SBIC into MBIC by level information. Assume that the band was $b(l_1, l_2, \dots, l_{n-1}, l_n)$, where $l_1, l_2 \dots, l_{n-1}, l_n$ were the different components of the band. An m-bit integer SC is used to represent a fixed-scale band (the integer types in computers are 32-bit and 64-bit). The SC is transformed into the integer coding MC of different levels according to the level information.

Since the band span is from kilometer to picometer, a 64-bit integer was used to represent single-scale band coding. Let the band be $b(l_1, l_2, l_3, l_4, l_5, l_6, l_7, l_8)$, where the memory usage of the components of the band is as follows:

1. The range of $l_8$-pm is 0–1000, represented by a 10-bit binary, where 1000–1023 is a null value;
2. The range of $l_7$-nm is 0–1000, represented by a 10-bit binary number, where 1000–1023 is null;
3. The range of $l_6$-µm is 0–1000, represented by a 10-bit binary number, where 1000–1023 is null;
4. The range of $l_5$-mm is 0–10, represented by a 4-bit binary number, where 10–16 is null;

5.  The range of $l_4$-cm is 0–1000, represented by a 4-bit binary number, where 10–16 is null;
6.  The range of $l_3$-dm is 0–1000, represented by a 4-bit binary number, where 10–16 is null;
7.  The range of $l_2$-m is 0–1000, represented by a 10-bit binary number, where 10–16 is null;
8.  $l_1$-km is represented by a 12-bit binary number.

For example, 1 pm is the fixed scale. The *SC* is made up of $l_1$(12-bit), $l_2$(10-bit), $l_3$(4-bit), $l_4$(4-bit), $l_5$(4-bit), $l_6$(10-bit), $l_7$(10-bit), and $l_8$(10-bit) in memory. As shown in Figure 4, the band range is 0–4096 km, denoted by integers ranging from 0 to $2^{64}$-1. Since the commonly used scales (km, m, dm, ... , nm, pm) are not integral multiples of 2, SC is not continuous.

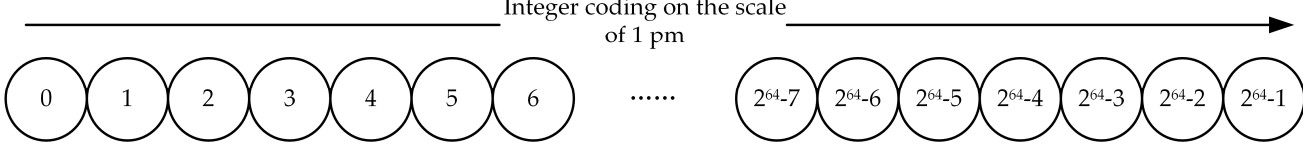

**Figure 4.** Integer coding at 1 pm scale.

Since SBIC already occupies almost all 64-bit integers, it is necessary to select some integers from them to represent other scale bands. We chose 1-bit from 64-bit to store multi-scale band integer encoding. In this way, the single-scale band integer at the 1 pm scale changed from 0~$2^{64}$-1 to 0~$2^{63}$-1, indicating that the range was 0~2048 km, and the remaining $2^{63}$ integers were used to store bands of other scales. The $2^{64}$ integers were divided into 64 levels according to the structure of the binary tree, effectively including the commonly used units of length (km, m, dm, ... , nm, pm), where level 63 consisted of $2^{63}$ integers, level 63 consisted of by $2^{62}$ integers, ... , level 0 was represented by 1 integer, the minimum scale level was 63, and the relationship between adjacent scales was a factor of 2. The correspondence between levels and scales is shown in Table 1.

**Table 1.** Corresponding levels of different scales.

| Level | Scale | Level | Scale | Level | Scale | Level | Scale |
|-------|-------|-------|-------|-------|-------|-------|-------|
| 63 | 1 pm | 47 | 64 | 31 | 4 | 15 | 64 |
| 62 | 2 | 46 | 128 | 30 | 8 | 14 | 128 |
| 61 | 4 | 45 | 256 | 29 | 1 cm | 13 | 256 |
| 60 | 8 | 44 | 512 | 28 | 2 | 12 | 512 |
| 59 | 16 | 43 | 1 µm | 27 | 4 | 11 | 1 km |
| 58 | 32 | 42 | 2 | 26 | 8 | 10 | 2 |
| 57 | 64 | 41 | 4 | 25 | 1 dm | 9 | 4 |
| 56 | 128 | 40 | 8 | 24 | 2 | 8 | 8 |
| 55 | 256 | 39 | 16 | 23 | 4 | 7 | 16 |
| 54 | 512 | 38 | 32 | 22 | 8 | 6 | 32 |
| 53 | 1 nm | 37 | 64 | 21 | 1 m | 5 | 64 |
| 52 | 2 | 36 | 128 | 20 | 2 | 4 | 128 |
| 51 | 4 | 35 | 256 | 19 | 4 | 3 | 256 |
| 50 | 8 | 34 | 512 | 18 | 8 | 2 | 512 |
| 49 | 16 | 33 | 1 mm | 17 | 16 | 1 | 1024 |
| 48 | 32 | 32 | 2 | 16 | 32 | 0 | 2048 |

As shown in Table 1, 64 scales are represented by 64-bit integers, namely: 1 pm, 2 pm, ... , 1 nm, 2 nm, ... , 1 µm, 2 µm, ... , 1 mm, 2 mm, ... , 1 cm, 2 cm, ... , 1 dm, 2 dm, ... , 1 m, 2 m, ... , 1 km, 2 km, ... , 2048 km, with scales ranging from 1 pm to 2048 km. To include the common scale of the band, 1 nm is extended to 1024 pm, 1 µm to 1024 nm, 1 mm to 1024 µm, 1 cm to 16 mm, 1 dm to 16 cm, 1 m to 16 dm, 1 km to 1024 m. As shown in Figure 5, a 64-layer binary tree structure was obtained.

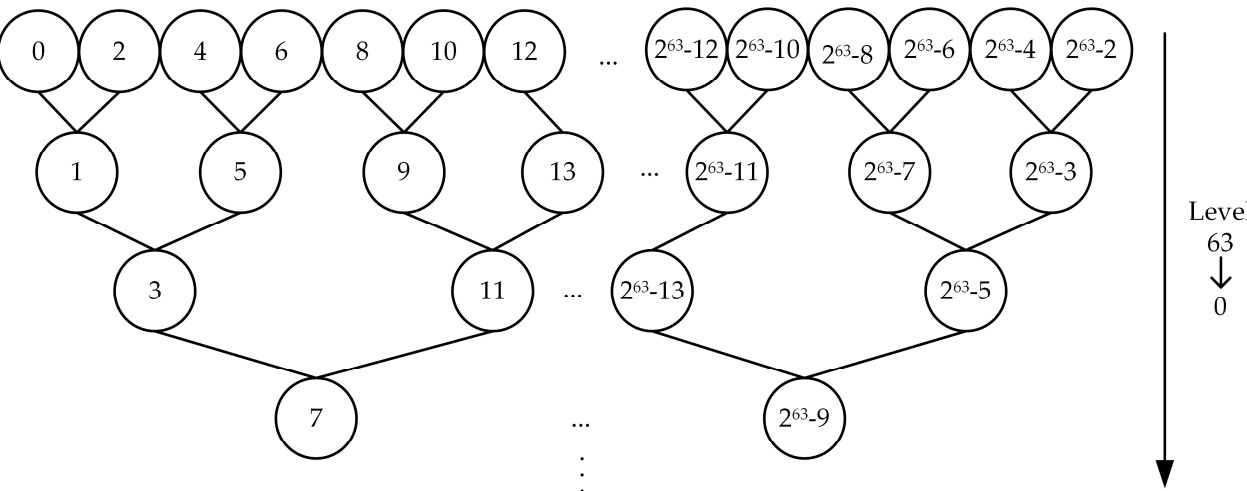

**Figure 5.** Multi-scale band integer coding.

The MBIC is obtained by the level $N$ and $b(l_1, l_2, l_3, l_4, l_5, l_6, l_7, l_8)$, and the specific method is as follows:

1.  Single-scale band integer coding calculation: SC is calculated according to Formula (7);

$$SC = (l_1 << 52)(l_2 << 42)(l_3 << 38)(l_4 << 34)(l_5 << 30)(l_6 << 20)(l_7 << 10)l_8 \quad (7)$$

2.  Multi-scale band integer coding calculation: according to Formulas (8)–(10), the multi-scale band integer coding mc is obtained by using the level $N$;

$$SC = SC << 1 \quad (8)$$

$$Deta0 = 1 << (63 - N) \quad (9)$$

$$MC = (sc >> (64 - N)) << (64 - N) + Deta0 - 1 \quad (10)$$

where *Deta0* is the smallest number in the Nth level.

### 2.3.2. MBIC Related Operations

Since MBIC represents band data by integers, the related operations in MBIC mainly involve the addition and subtraction of integers and bit operations. This section introduces the level calculation and relationship calculation method of MBIC in detail.

#### Level Calculation

The multi-scale band integer code is a 64-bit integer, so the level information cannot be intuitively obtained by giving the integer. It is necessary to calculate its level. According to the parity of *MC*, the specific methods are as follows:

1.  If MC is an even number, its level $N$ is 63;
2.  If MC is an odd number, first, calculate how much the high-order bits in front of the binary of *MC − 1* and *MC + 1* are the same, i.e., *Mid = (MC − 1) ˆ (MC + 1)*. Secondly, the level is calculated by calculating how many consecutive zeros are on the left side of the binary of *Mid*. MBIC is represented by a 64-bit integer and can use the bifurcation method to efficiently obtain level information. The branch method judges how many 0 are on the left of the 64-bit integer according to the method of dichotomy.

Level Relationship Calculation

The multi-scale band integer encoding has a containment relationship and a contained relationship. The child coding set can be obtained by using the containment relationship, and the parent coding set can be obtained by the contained relationship.

1.  Child coding set: Given a multi-scale band integer encoding *MC*, the corresponding level is *N*. The integer encoding *MC'* of the calculated level *N'* ($N' \geq N$) is the child coding set. Let the interval of the child coding set be [$C_1$, $C_2$], where $C_1$, $C_2$ are calculated as Formulas (11) and (12):

$$C_1 = MC - (1 << (63 - N)) + 1 \tag{11}$$

$$C_2 = MC + (1 << (63 - N)) + 1 \tag{12}$$

2.  Parent coding set: Let the *MC* level be *N*, and the parent encoding level is *N'*. The integer *MC'* of the calculated level *N'* ($N' < N$) is the parent coding set. According to Formulas (13) and (14), the parent coding set of *MC* is obtained from *N − 1* to 0 through loop variable *N'*:

$$Deta0 = 1 << (63 - N') \tag{13}$$

$$FMC = (MC >> (64 - N') << (64 - N')) + Deta1 - 1 \tag{14}$$

### 2.3.3. The Association Method between MBIC and Band

The bands often exist in the form of an interval, and establishing the association between band intervals and MBIC is crucial for data retrieval. Since MBIC is designed according to the binary tree rules based on common granularity units, the following rules are designed to establish the association between band and MBIC:

- Rule 1: The maximum level $N_{max}$ of MBIC is not larger than the maximum level $N_{max}'$ of the start and end point of the band.
- Rule 2: First, the bands are padded with fine-grained to coarse-grained integer encoding, then the bands are padded with coarse-grained to fine-grained integer encoding until the band interval is filled. The specific filling method is shown in Figure 6, where L represents the band, and A, B, C, and D represent multi-scale integer coding at different levels.

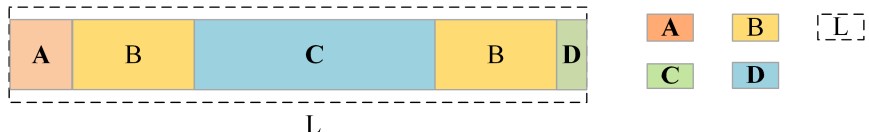

**Figure 6.** An example of padding from fine-grained to coarse-grained.

The steps to associate the band with the multi-scale band integer coding are as follows:

1.  Convert the start and end point of the bands to the same granularity.

Analyze the levels of the start($b_1(l_i)$) and end ($b_2(l_j)$) points of the bands. If $i \neq j$, use the conversion function to convert coarse-grained to fine-grained. When the granularity of $b_1$ is coarser than that of $b_2$, the $T_s$ conversion function is used, and when the granularity of $b_1$ is finer than that of $b_2$, the T1 conversion function is used;

2.  Gradually divide and determine its level scope.

Assuming that both $b_1(l_i)$ and $b_2(l_j)$ are data at the micrometer scale, i.e., *i = j = 6*, according to each component, its grade is divided into 6 grades (33~43, 29~33, 25~29, 21~25, 11~21, 0~11). The minimum level $N_{min}$ and the maximum level $N_{max}$ of the MBIC

is determined grade by grade. The maximum level is the maximum level at this grade, i.e., $N_{max}$ = 43 at the (33~43) grade. The minimum level calculation is divided into two cases:

- Case 1: If $l_{i-1} = l_{j-1}$, calculate the band length $l$, i.e., $l = l_j - l_i + 1$, and convert $l$ to the sum of the power of 2, where the maximum value in the addend corresponds to the level of is $N_{min}$;
- Case 2: If $l_{i-1} \neq l_{j-1}$, calculate the band length $l$, $l = max_j - l_i + 1$, where $max_j$ is the maximum value of the $j$-th component, for example, if $j = 6$, then $max_j = 1000$. Then convert $l$ to the sum of the power of 2, where the maximum value in the addend corresponds to the level of is $N_{min}$;

3.  Accurate filling step by step.

According to Table 1, obtain the level $N$ of the corresponding component for each grade, if $N \leq N_{min}$, convert the $l$ of this grade to the sum of the power of 2, and obtain the level corresponding to the addend. Finally, calculate the multi-scale band integer encoding according to the level information; if $N > N_{min}$, execute the loop body until $l = 0$. Assuming that the corresponding scale of $N$ is $v$, the loop body is as follows:

$l = l - v$. If $l > 0$, multi-scale integer encoding is performed on the data of the current level and $N = N - 1$, $l_i = l_i + v$; If $l < 0$, $N = N + 1$, $l = l + v$; If $l = 0$, multi-scale integer encoding is performed on the data at the current level and the loop is exited.

For example, the band range is (6 km 626 m 4 dm 5 cm 1 mm~6 km 626 m 4 dm 5 cm 4 mm).

1.  Step 1: Calculate the corresponding level of $b_1$ and $b_2$, $N_1 = 33$, $N_2 = 33$;
2.  Step 2: According to the components of $b_1$ and $b_2$, it is divided into 5 grades (29~33, 25~29, 21~25, 11~21, 0~11); It is only necessary to calculate the band length $l$ at the (29~33) grade, $l = 4$ mm, the level corresponding to 4 mm is $N_{min} = 27$, $N_{max} = 33$;
3.  Step 3: The level corresponding to $l_5 = 1$ mm is $N = 33$, $N > N_{min}$, and the multi-scale integer coding is obtained: $MC_1$= 59,551,923,803,521,023 ($N = 33$), $MC_2$= 59,551,927,024,746,495 ($N = 32$), $MC_3$= 59,551,930,245,971,967 ($N = 33$);

As shown in Figure 7, the relationship between MBIC and band is many-to-many.

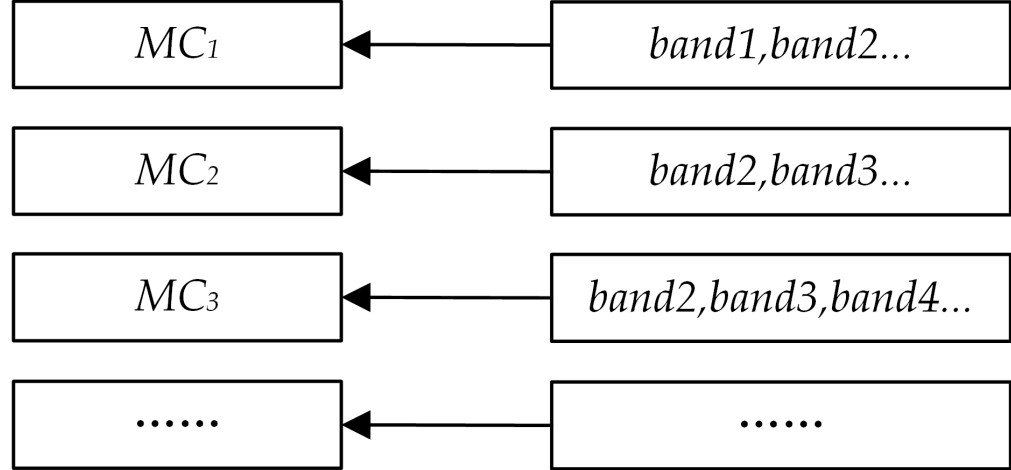

**Figure 7.** Correlation between MBIC and bands.

### 3. Results

To verify the effectiveness of the design method in this paper, we conducted related experiments on multi-granularity dimensions (time, band) that satisfy the fuzziness of dimension granularity. The verification content mainly includes the following three points: the effectiveness of the DGFQM, the relevant factors that affect the query efficiency of MTSIC and string coding, and the influence of the association method between MBIC and

band on data retrieval. In response to the above contents, we designed the experiments as follows:

Experiment 1: To verify the effectiveness of the DGFQM, we simulated time data, and then compared the query results of the DGFQM and the intersection query method.

Experiment 2: We designed time data sets with different proportions using string encoding and MTSIC methods and compared the retrieval efficiency of the two ways.

Experiment 3: We used the string coding method and the association method between MBIC and band to build an index table for the simulated band data, respectively, and then compared the query efficiency of the two ways.

Development experiment environment: Windows Intel(R) Core(TM) i5-8500 CPU @ 3.00 GHz, 64-bit,8 GB, Visual Studio 2019, C++, MySQL 5.7.19.

### 3.1. DGFQM

At present, we mostly use the intersection query method for data queries. We used string coding and MTSIC to store time data, respectively, and then compared the results of the DGFQM and the intersection query method. First, randomly generate n different time scales (year, month, day, hour, minute, second, millisecond, microsecond), then perform string coding and multi-scale integer coding. Finally, build a B-tree for the intersection query method and the DGFQM.

### 3.1.1. The DGFQM Based on String Coding

Dimension granularity fuzziness query steps based on string coding:

1. Perform string encoding on the query interval $[t_1, t_2]$ to obtain the string interval $[s_1, s_2]$;
2. Decode the strings $s_1$ and $s_2$ to attain levels $N_1$, $N_2$;
3. Parse the string $s_1$, and then obtain the parent data set $C_{f1}$ of $s_1$ by coding;
4. Parse $s_2$, and then obtain the child set $C_{s2}$ of $s_2$ through string coding;
5. Obtain query results through set operations and query statements;

Set n to be 10,000, 100,000, 500,000, 1,000,000, 5,000,000, 10,000,000, and select various query intervals to perform the intersection query and the DGFQM, respectively. The query intervals are the annual scale, the daily scale, and the second scale. The query results are shown in Figure 8. The intersection query method does not take into account the dimension granularity fuzziness, but only relies on the size sorting function of the code to obtain the data. Therefore, the number of results obtained by DGFQM is higher than that of the intersecting query method. From Figure 8, it can be seen that the amount of missing data in the intersection query is affected by the amount of data and the query interval. The amount of missing data is proportional to the query interval and the total amount of data.

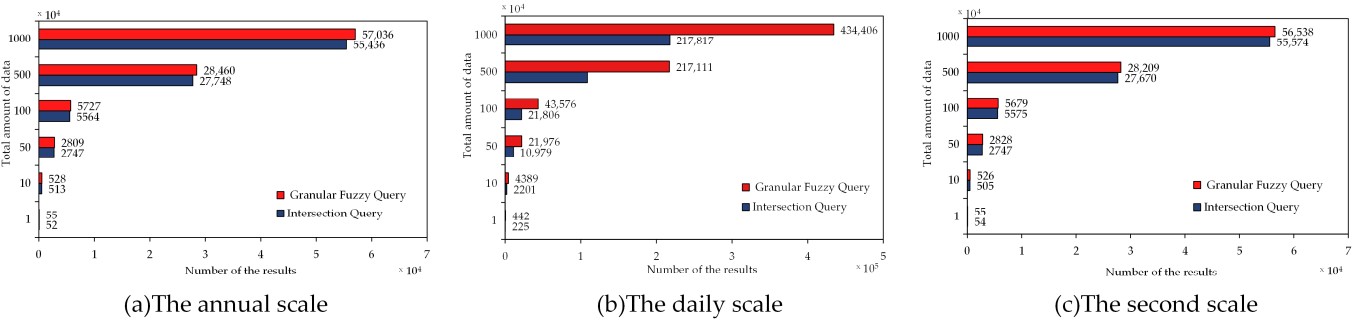

(a)The annual scale        (b)The daily scale        (c)The second scale

**Figure 8.** The number of query results for both methods. (**a**) the number of query results for the annual scale query interval. (**b**) the number of query results for the daily scale query interval. (**c**) the number of query results for the second scale query interval.

To verify the correctness of the data in the query results of the dimension granularity fuzziness, we took the query interval (15 November 2014, 15 February 2015) as an example to compare query results for both methods under the 1 million data set. The number of query results for the DGFM is 5727, and the number of unequal results is 5564. The query results of the intersection query method are 5564, of which 5408 are unique. As shown in Table 2, the query results of the DGFM are more complete than the intersection query.

**Table 2.** Two query results based on the string encoding.

| Partial Results of Granular Fuzzy Queries | Partial Results of an Intersect Query | Partially Missing Data for Intersecting Queries |
|---|---|---|
| '2014'<br>'2014-11'<br>'2014-11-15'<br>'2014-11-15T00:08:08.216495'<br>'2014-11-15T01:25'<br>'2014-11-15T01:59:09.074094'<br>'2014-11-15T03:08:31.252138'<br>'2015-02-15T00:10:09.460989'<br>'2015-02-15T00:21:15.373' | '2014-11-15'<br>'2014-11-15T00:08:08.216495'<br>'2014-11-15T01:25'<br>'2014-11-15T01:59:09.074094'<br>'2014-11-15T03:08:31.252138' | '2014'<br>'2014-11'<br>'2015-02-15T00:10:09.460989'<br>'2015-02-15T00:21:15.373' |

### 3.1.2. The DGFQM Based on MTSIC

The DGFQM steps based on MTSIC:

1. According to the multi-scale time segment integer encoding method, the integer coding $MTC_1$ and $MTC_2$ of $t_1$ and $t_2$ were obtained, so the integer coding interval was $C_b = [MTC_1, MTC_2]$;
2. Calculate the level of $MTC_1$ and $MTC_2$, and obtain the corresponding levels $N_1$ and $N_2$ through level operations;
3. The parent data sets $C_{f1}$ and $C_{f2}$ are obtained through the contained relationship operation, and the missing fuzzy data set $C_1$ is obtained according to Formula (15);

$$C_1 = \left\{ x \middle| x \in C_{f1} \vee x \in C_{f2} \wedge x \notin C_b \right\} \tag{15}$$

4. The child data sets $C_{s1}$ and $C_{s2}$ of $MTC_1$ and $MTC_2$ were obtained by using the containment relationship operation, and then the missing precise data set $C_2$ was obtained according to the following Formula (16);

$$C_2 = \left\{ x \middle| x \in C_{s1} \vee x \in C_{s2} \wedge x \notin C_b \right\} \tag{16}$$

5. Obtain query results through set operations and query statements;

Set n to be 10,000, 100,000, 500,000, 1,000,000, 5,000,000, 10,000,000, and select various query intervals to perform the intersection query and the DGFQM respectively. The query intervals are the annual scale, the daily scale, and the second scale. The query results were consistent with the query result based on string coding, as shown in Figure 8.

### 3.2. The Influence of the Proportion of Different Time Scales on Retrieval Efficiency

MTSIC uses an integer type to store time data, which occupies less memory and is more computationally efficient than a string type. Therefore, the proportion of different scales in the time data may have an impact on the query efficiency. We designed different temporal data sets to compare the query efficiency of temporal string encoding and MTSIC using DGFQM. The experimental design process was as follows:

1.  Randomly generate n time data (year, month, day, hour, minute, second, millisecond, microsecond) according to equal and unequal proportions. The non-proportional data is generated in the way of 1: 2: 4: 8: 16: 32: 64: 128, which will generate a combination of factorials of 8, so we divided the scales into fine scales (hour, minute, second, millisecond, microsecond) and coarse scales (year, month, day). The specific design is shown in Table 3.
2.  Establish a B-tree index. Perform string encoding and MTSIC on time data, and then build B-trees, respectively.
3.  Dimension granularity fuzzy query. According to Section 3.1, we performed the DGFQM on string coding and MTSIC, respectively, and counted the results.

**Table 3.** Proportion designs in the temporal data set.

| Proportional Way | Representation Symbols | Proportional Design |
| --- | --- | --- |
| y: m: d: h: m: s: ms: μs | *dbl* (equal proportion) | 1: 1: 1: 1: 1: 1: 1: 1 |
| | *bdbl* (unequal proportion) | 1: 2: 4: 8: 16: 32: 64: 128 |
| | *fdbl* (unequal proportion) | 128: 64: 32: 16: 8: 4: 2: 1 |

Set n to 10,000, 10,000, 100,000, 1,000,000, 5,000,000, 10,000,000, and select the query range: "2014 to 2015", "15 November 2014 to 15 February 2015" for querying. Each query result was taken ten times, and the query efficiency was counted. The result was shown in Figure 9.

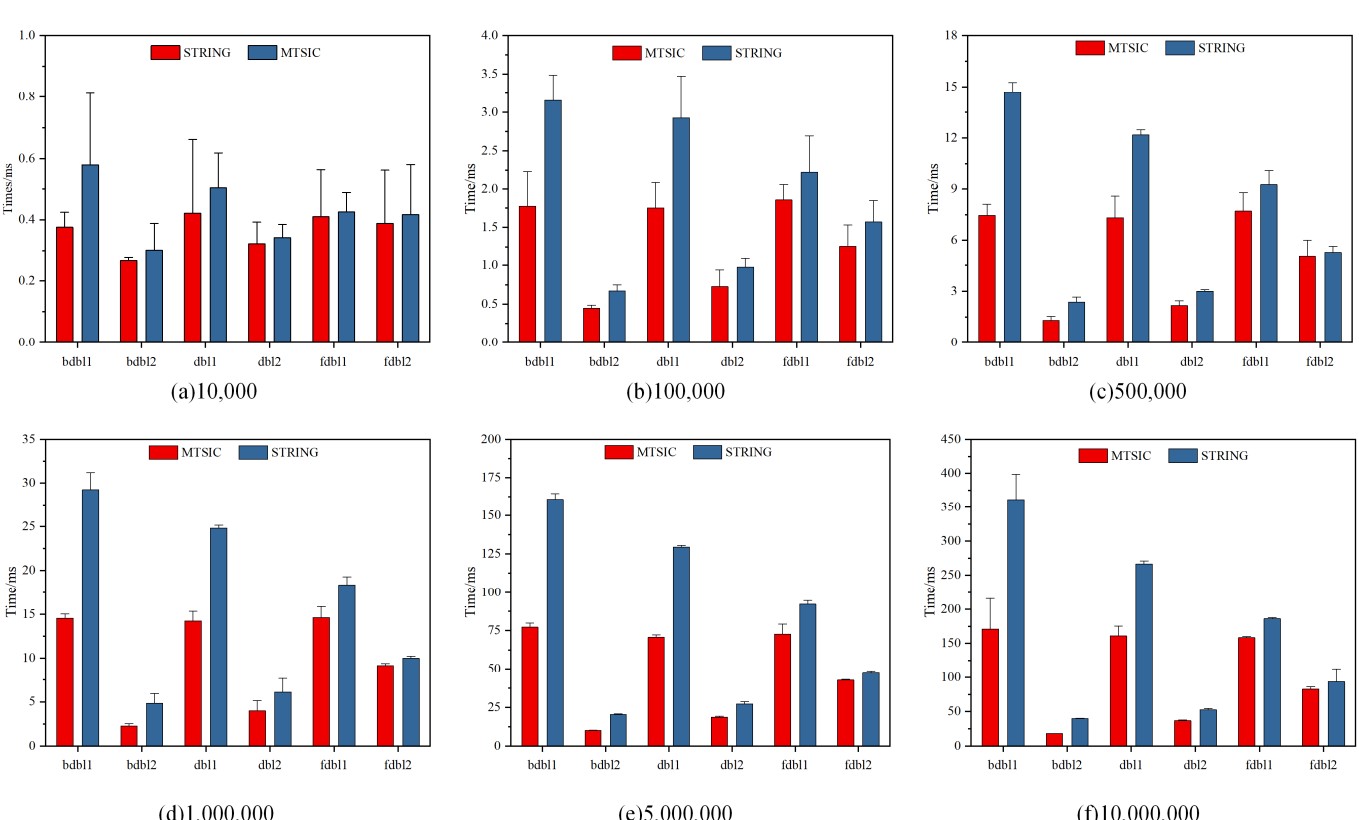

**Figure 9.** The query time of the two coding methods under different time data (1: "2014–2015"; 2: "15 November 2014–15 February 2015").

The red marks in Figure 9a–e are all lower than the blue marks, so the query time of MTSIC is less than that of string coding. Under the data volume of 10,000, the time-consuming of the string coding was 1.2 (*dbl1*) times, 1.5 times (*bdbl1*), 1.1 times (*dbl2*), and 1.2 times (*bdbl2*) of MTSIC, respectively. Under the 10,000 data volume of

*fdbl*, the time-consuming of string coding was roughly equal to that of MTSIC. The time-consuming of string coding under 10 million data volume is 1.2 times (fdbl1), 1.7 times (*dbl1*), 2.1 times (*bdbl1*), 1.1 times (*fdbl2*), 1.5 times (*dbl2*), 2.1 times (*bdbl2*) of MTSIC, respectively. Therefore, we can draw the following conclusions: Under the same proportion, with the increase in the total amount of data or the expansion of the query scope, the query effect of MTSIC was better and better compared to string coding. In the case of the same amount of data, with the increase in the fine-scale ratio, the query effect of MTSIC was better and better.

### 3.3. Comparing the Retrieval Efficiency of MBIC and String Encoding

Randomly generate n bands and manage them in two ways. One was the string coding method, which was stored and indexed through two fields of string type. The other was to use the association method between MBIC and band to store and index. Table 4 is a comparison of the expressions of the two codes. Let the band be $[b_1, b_2]$, and retrieve data according to the DGFQM. The steps for the DGFQM of bands were as follows:

**Table 4.** Comparison of two coding methods.

| Storage Method | Method Description | Example |
|---|---|---|
| string | Use two fields to store bands | "6-626-4-5-1"–"6-626-4-5-4" |
| MBIC | Store bands with a column of integer | The multi-scale integer encoding of "6-626-4-5-1"–"6-626-4-5-4" is: 59,551,923,803,521,023, 59,551,927,024,746,495, 59,551,930,245,971,967 |

The steps of the DGFQM based on string coding:

1. Perform string coding on the query interval $[b_1, b_2]$ to obtain the string interval $[s_1, s_2]$;
2. Attain the exact data set $C_x$ in the query interval. Let the storage fields be *field₁* and *field₂*, respectively, and obtain the exact data set $C_x$ according to Formula (17);

$$C_x = \{filed_1 \leq s_1 \leq filed_2 \vee filed_1 \leq s_2 \leq filed_2\} \tag{17}$$

3. Obtain the fuzzy data set $C_m$ in the query interval. Obtain the fuzzy data set $C_m$ according to the Formula (18);

$$C_m = \{s_1 \geq filed_1 \wedge s_2 \geq filed_2\} \tag{18}$$

4. Obtain query results through set sum operation;

The steps of the DGFQM based on MBIC:

1. According to the association method between MBIC and band, the corresponding MBIC set $B = \{MC_1, MC_2,..., MC_n\}$ is obtained;
2. Attain the exact data set $C_x$ in the query interval. Obtain the child interval $x_i$ of the *i*-th code in B by including relational operation, i.e., $B(i)$ and repeat the operation until all codes in *B* are traversed. The specific process was shown in Figure 10a:
3. Attain fuzzy data set $C_m$ of query interval. Obtain the parent interval $m_i$ of the *i*-th code in B by including relational operation, i.e., $B(i)$ and repeat the operation until all codes in B are traversed. The specific process was shown in Figure 10b;
4. Obtain query results through set operations;

Set n to 500,000, 1,000,000, 5,000,000, and 10,000,000, and make multiple queries. We considered four query intervals as an example, which contained four different scale intervals. The query intervals were represented by the string coding method and the multi-scale integer coding method, and the specific design is shown in Table 5. Then query according to DGFQM under different codes. Finally, take ten times for each query and count the query efficiency.

$$\{MC_1,\ MC_2,\ MC_3,\ \cdots,\ MC_n\} \xrightarrow[\text{relationship}]{\text{containment}} x_1,\ x_2,\ x_3,\ \cdots,\ x_i \xrightarrow{\text{set operations}} C_x$$

<center>(a)accurate data acquisition process</center>

$$\{MC_1,\ MC_2,\ MC_3,\ \cdots,\ MC_n\} \xrightarrow[\text{relationship}]{\text{contained}} m_1,\ m_2,\ m_3,\ \cdots,\ m_i \xrightarrow{\text{set operations}} C_m$$

<center>(b)fuzzy data acquisition process</center>

**Figure 10.** Data acquisition process.

**Table 5.** Corresponding codes for different queries.

| | Query Interval | MBIC | String Coding |
|---|---|---|---|
| query1 | 4,003,612~4,003,619 mm | 36,058,524,635,103,231<br>36,058,531,077,554,175<br>36,058,537,520,005,119 | "04-003-6-1-2"–"04-003-6-1-9" |
| query2 | 400,362~400,367 cm | 36,058,586,912,129,023<br>36,058,689,991,344,127 | "04-003-6-2"–"04-003-6-7" |
| query3 | 40,032~40,039 dm | 36,056,834,565,472,255<br>36,058,483,832,913,919<br>36,060,133,100,355,583 | "04-003-2"–"04-003-9" |
| query4 | 2004~2060 m | 18,067,175,067,615,231<br>18,119,951,625,748,479<br>18,225,504,742,014,975<br>18,366,242,230,370,303<br>18,471,795,346,636,799<br>18,524,571,904,770,047<br>18,546,562,137,325,567 | "02-004"–"02-060" |
| query5 | 4003~4230 m | 36,059,583,344,541,695<br>36,072,777,484,075,007<br>36,081,573,577,097,215<br>36,134,350,135,230,463<br>36,239,903,251,496,959<br>36,451,009,484,029,951<br>36,873,221,949,095,935<br>37,436,171,902,517,247<br>37,858,384,367,583,231<br>38,016,714,041,982,975<br>38,056,296,460,582,911 | "04-003"–"04-230" |

The statistical results are shown in Figure 11. The association method between MBIC and band proposed in this paper has a better effect than the traditional string representation. The query time for both methods increase with the amount of data. Under the same amount of data, when using the method proposed in this paper, the query time gradually increased with the expansion of the band range. It can be seen from Figure 11 that the time-consuming of queries 1–3 was about zero. However, when using the string coding method to retrieve the band range, it is necessary to traverse all the data, which took a long time. The results show that the query band range has little effect on it.

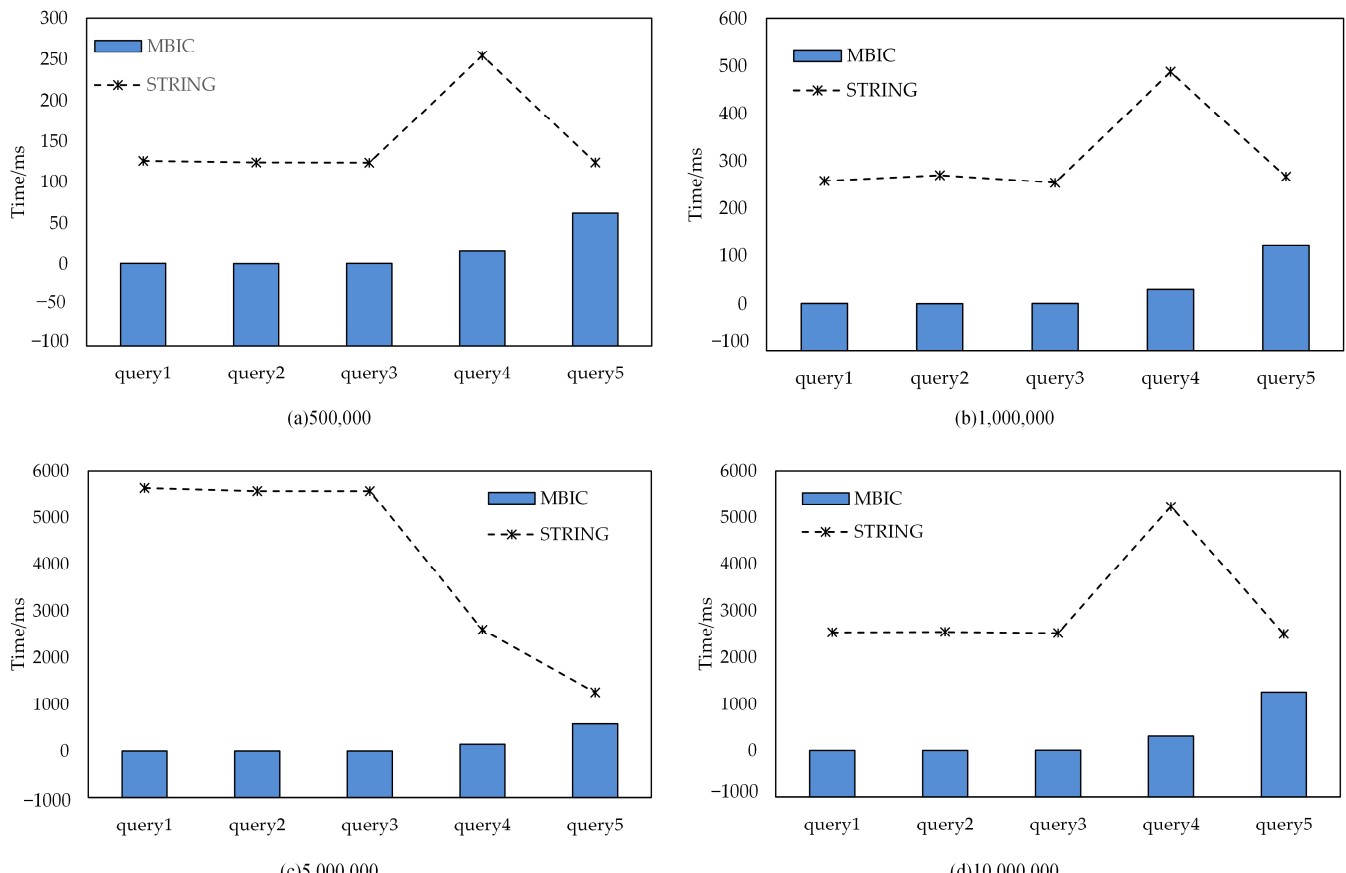

**Figure 11.** The query time of the two coding methods under different band data.

### 3.4. Discussion

Aiming at the problem of the multi-granularity dimension in spatiotemporal data, we proposed a management method of multi-granularity dimensions for spatiotemporal data. Mainly study the fuzziness and organization methods of multi-granularity dimensions. First, according to the inclusion relationship between granularities, we proposed DGFQM, which solved the problem of data loss caused by the multi-granularity characteristic of dimensions. Second, we discussed the encoding method of bands and designed the association method of multi-scale integer coding and bands. The correlation experiments were carried out by simulating time and band data. Correlation experiments are carried out by simulating time and band data. The experimental results are as follows:

(1) Whether the string coding method or MTSIC, the DGFQM can obtain more complete data than the intersection query method;

(2) Although the query efficiency of MTSIC is higher than that of the string coding method, its effect is affected by the proportion of different scales in the data. With the increase in the amount of fine-scale data, the query effect of multi-scale time integer coding is better;

(3) Compared with the string coding method, the association method between MBIC and band designed in this paper effectively improves the data retrieval efficiency. The retrieval efficiency of this method is related to the range of the query band, and the query effect is better as the range of the band decreases. Especially when the band range is small, the query time is about 0.

### 4. Conclusions

#### 4.1. DGFQM

Few studies have discussed the fuzziness caused by the multi-granularity of dimensions. Although a cross-scale spatial filling curve was proposed in reference [16] to provide

a query method for multi-scale spatial data, the relevant theories and methods of dimension granularity fuzzy such as time were not proposed. In this paper, we discuss the fuzziness of multi-granularity dimensions from point and segment, and proposed the DGFQM. To verify the effectiveness of the DGFQM, we simulated temporal data and compared the query results of the intersection query method [25] and DGFQM.

### 4.2. Multi-Scale Integer Coding

At present, multi-scale integer coding has achieved good results in time and space. However, there were few studies on other multi-granularity dimensions. The concept of time-spectrum was proposed in reference [34], which put our focus on spectral information. We extended multi-scale integer coding to multi-scale dimension and took the band as an example to describe the application of multi-scale integer coding in a band in detail. We used the scale information contained in multiscale integer coding to design the correlation method between multiscale integer coding and band. The band was converted into a one-dimensional array by filling. The experiment showed that the association method proposed in this paper improved the efficiency of data retrieval compared with the traditional binary form.

In the above research, we studied the multi-granularity metric in spatiotemporal data from the above two aspects. The results were generally good, but there were still some limitations, and there are still some problems to be discussed.

(1) This method was to solve the problem of incomplete query results based on time and other multi-scale dimensions. This requires that the query data cover as many areas as possible. Secondly, the method uses multi-scale integers to fill multi-scale dimensions. When the scale is one year, three months, one day, and five hours, this complex situation needs to be filled with many multi-scale integer codes, which would affect the efficiency of data retrieval.

(2) We analyzed the fuzziness of spatiotemporal data from the multi-scale dimension level, and provided a new perspective for the study of spatiotemporal data fuzziness. We obtained fuzzy data with hidden values from the data through the DGFQM, so as to better understand and analyze the change trend in various fields such as economy and culture. Next, we will further study the query results, analyze the potential information in the fuzzy data, and build the corresponding knowledge map.

(3) We applied multi-scale integer coding to the band, and discussed the applicability of multi-scale integer coding. It can be seen that multi-scale integer coding has certain advantages in terms of memory occupation and query efficiency. At present, multi-scale integer coding was applied to time, space, and band, respectively. Next, we will consider building the coding of a space-time, spatiotemporal spectrum based on multi-scale integer coding.

**Author Contributions:** Conceptualization, Wen Cao and Wenhao Liu; methodology, Wen Cao, Wenhao Liu and Xiaochong Tong; software, Wenhao Liu; validation, Wenhao Liu, Jianfei Wang, Feilin Peng, Yuzhen Tian, Jingwen Zhu; formal analysis, Wenhao Liu, Feilin Peng, Yuzhen Tian and Jingwen Zhu; investigation, Wenhao Liu; resources, Wenhao Liu; data curation, Wenhao Liu; writing—original draft preparation, Wen Cao and Wenhao Liu; writing—review and editing, Wen Cao, Wenhao Liu and Jianfei Wang; visualization, Wenhao Liu; supervision, Wen Cao; project administration, Wen Cao and Wenhao Liu; funding acquisition, Wen Cao. All authors have read and agreed to the published version of the manuscript.

**Funding:** This work was supported by The Excellent Youth Foundation of Henan Municipal Natural Science Foundation (212300410096), Program of Song Shan Laboratory (Included in the Management of Major Science and Technology Program of Henan Province) under Grant number 221100211000-03, and The National Key R&D Plan of China (2018YFB0505304).

**Institutional Review Board Statement:** Not applicable.

**Informed Consent Statement:** Not applicable.

**Data Availability Statement:** Not applicable.

**Conflicts of Interest:** The authors declare no conflict of interest.

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
