# Peer review of "A Management Method of Multi-Granularity Dimensions for Spatiotemporal Data"

_ijgi, doi:10.3390/ijgi12040148_

Round 1

Reviewer 1 Report

Please attachment for comments.

Author Response

Dear Reviewer1:

Thank you for your comments concerning our manuscript entitled “A Management Method of Multi - granularity Dimensions for Spatiotemporal Data” (ID: ijgi-2136115). Those comments are all valuable and very helpful for revising and improving our paper, as well as the important guiding significance to our researches. We have studied comments carefully and have made corrections which we hope meet with approval. Revised portions are marked up using the “Track Changes” function in this manuscript. The main corrections in the paper and the responses to the reviewers’ comments can be found below. The revised manuscript is uploaded to the Submit Revised Manuscript section.

Reviewer 2 Report

A Management Method of Multi-granularity Dimensions for Spatiotemporal Data

Managing multigranularity dimensional information and quickly obtaining multi-dimensional information required for application analysis is an urgent problem to be solved at present. In this manuscript, the authors propose a management method of multi-granularity dimensions for spatiotemporal data. First, considering that the multi-granularity of dimensions leads to different results when describing data, they design a proper method so-called the Dimension Granularity Fuzzy Query Method (DGFQM). This method divides the query results according to the granularity and obtains the complete data set through the relevance of different granularity. Second, they simulated time and waveband to conduct experiments. The experimental results show that: (1) The DGFQM obtains complete data; (2) For temporal data, although multi-scale integer encoding is better than string encoding, the optimization effect is affected by the proportion of different time scales. (3) Multi-scale integer coding is also applicable to band attributes.

The topic shown in this manuscript is interesting and the research is well presented. I can recommend this manuscript for publication after the following suggestions are addressed:

1.       I am missing a first paragraph where the authors introduce a brief context about the problem. Why is it so important and relevant for being studied?

2.       In the introduction the authors argue: “Spatiotemporal data contains a wealth of information and plays an important role in scientific decision-making, government management, and public services. Depending on the depth of its application, it is divided into descriptive analysis application, predictive analysis application, and directive analysis application [1,2].” I agree, and even this paragraph can be part of the context. However, it is a bit ambiguous and too generic to introduce your research in this way. Let me guide you at this point with some questions_: Why do we need more data?? Why do we need to invest more and more in data? -> Basically, we are part of a knowledge society and most of the problems we face must be addressed based on data-driven approaches for understanding better and achieving more efficient and optimal decisions. In addition, the so called social space is very complex and all the human dynamics are over. To monitor and understand them is very complex and we need more and more data across scales. I recommend you two very recent studies where they analyze the continuous need of more data: “Scale, context and heterogeneity: the complexity of social space” and “The scales of human mobility”. I encourage to be more clear to introduce your study.

3.       The authors argue: “Spatiotemporal data big data contains multi-dimensional information, in which dimensions such as time and bands have multi-granularity characteristics, which brings a great challenge to data management.” (a) The expression “Spatiotemporal data big data” sounds reiterative and not coherent. (b) “multi-dimensional information, in which dimensions such as time and bands” -> The dimensions of spatio-temporal information is expected to be space and time, not bands. (c) What the authors mean here “multi-granularity characteristics”? Do they refer to the different granularity that some datasets are collected? That is a key point for being introduced and well explained before. We have the availability to collect very fine-grained data nowadays (at individual level), but many times for protection issues ( for example, GDPR in Europe), this information cannot be distributed on this way. That means, there is a very sensitive aspect related to this. In the case of geographical information, this topic is especially sensitive. Some studies are focused on in this particular case by implementing Geospatial Analysis and Mapping Strategies for Fine-Grained and Detailed COVID-19 Data with GIS. I recommend you to address a review of this and to include some relevant studies related to this topic.

4.       Why the authors write “Spatiotemporal” in capital letter along the whole manuscript?

5.       In the introduction, the authors argue “At present, the effective method of Spatiotemporal data management was to establish a suitable Spatiotemporal data model for actual needs. However, the usage scenarios of Spatiotemporal data models were different, and the actual requirements were not unchanging, which increased the difficulty of matching the model with the actual requirements.” -> I cannot understand at all what the authors mean in this paragraph.

6.       However, with the rapid growth of Spatiotemporal data, the requirements for the storage capacity of databases are getting higher and higher.” -> I recommend you to check the study of Hidalgo (2016) “Why Information Grows”, where it is explained this point.

7.       After that, the authors argue: “In the above research, the existing database is used to manage the spatiotemporal data, and the spatiotemporal information of the spatiotemporal data is stored through the data type supported by the database.” The use of reiterations is excessive. Also the sentence remains pretty unclear.

8.       I have also observed that the resolution of some figures (such as the number 11) is very low.

9.       In the discussion, the authors end the manuscript with “We have studied the multi-granularity dimension in remote sensing data from the above two aspects. The results of the methods were good. Next, we will apply the research methods to specific management systems”. This paragraph needs to be expanded showing more clearly the results, but also the future perspectives to research.

Author Response

Dear Reviewerw2:

Thank you for your comments concerning our manuscript entitled “A Management Method of Multi - granularity Dimensions for Spatiotemporal Data” (ID: ijgi-2136115). Those comments are all valuable and very helpful for revising and improving our paper, as well as the important guiding significance to our researches. We have studied comments carefully and have made corrections which we hope meet with approval. Revised portions are marked up using the “Track Changes” function in this manuscript. The main corrections in the paper and the responses to the reviewers’ comments can be found below. The revised manuscript is uploaded to the Submit Revised Manuscript section.

Reviewer 3 Report

This is an interesting paper which has originality and appropriateness for this journal. In the Introduction chapter, it can be seen that the authors gave an overview of the current situation, but there is still a lot of literature that is not included in the list and more worldwide, which would still give a more precise picture of a better understanding of the spatiotemporal data management and multi-granularity metrics. Chapter material and methods is detailed with all required input data and well described methods. The results are precise and very clearly presented. The Discussion chapter is written as a conclusion. This part must either be corrected or changed.

Author Response

Dear Reviewerw3:

Thank you for your comments concerning our manuscript entitled “A Management Method of Multi - granularity Dimensions for Spatiotemporal Data” (ID: ijgi-2136115). Those comments are all valuable and very helpful for revising and improving our paper, as well as the important guiding significance to our researches. We have studied comments carefully and have made corrections which we hope meet with approval. Revised portions are marked up using the “Track Changes” function in this manuscript. The main corrections in the paper and the responses to the reviewers’ comments can be found below. The revised manuscript is uploaded to the Submit Revised Manuscript section.

Reviewer 4 Report

There are some problems in the manuscripts, such as:

(1) The expression of abstract should be modified carefully, so it should also be concluded again.

(2) The format of manuscript should be checked carefully, such as line 9-10 of Introduction.

(3) Please check the Figure 1(c), and determine it is Rx or Rx?

(4) Please check the Figure 3 and ensure the integrity of the picture.

(5) The most obvious is that the article is lack of conclusion, so the authors should add it.

(6) In addition, the contents and significance of the article need to be summarized more clearly. So, the creative points of this research can be seen by the readers.

Author Response

Dear Reviewerw4:

Thank you for your comments concerning our manuscript entitled “A Management Method of Multi - granularity Dimensions for Spatiotemporal Data” (ID: ijgi-2136115). Those comments are all valuable and very helpful for revising and improving our paper, as well as the important guiding significance to our researches. We have studied comments carefully and have made corrections which we hope meet with approval. Revised portions are marked up using the “Track Changes” function in this manuscript. The main corrections in the paper and the responses to the reviewers’ comments can be found below. The revised manuscript is uploaded to the Submit Revised Manuscript section.

Round 2

Reviewer 1 Report

The authors carefully considered the reviewer’s comments and made a large number of revisions, and the quality of the paper was greatly improved.

Reviewer 2 Report

The authors address most of my concerns highlighted in the previous review round. I realized the big effort by the authors for addressing them in the best way, meeting my satisfaction with the changes. I can now recommend this manuscript for publication.